# Listening to Caregivers’ Voices: The Informal Family Caregiver Burden of Caring for Chronically Ill Bedridden Elderly Patients

**DOI:** 10.3390/ijerph19010567

**Published:** 2022-01-05

**Authors:** Jinpitcha Mamom, Hanvedes Daovisan

**Affiliations:** 1Department of Adult Nursing and the Aged, Faculty of Nursing, Thammasat University, Khlong Luang, Pathum Thani 12121, Thailand; 2Excellence Center in Creative Engineering Design and Development, Faculty of Engineering, Thammasat University, Khlong Luang, Pathum Thani 12121, Thailand; 3Human Security and Equity Research Unit, Chulalongkorn University Social Research Institute, Chulalongkorn University, Bangkok 10330, Thailand; hanvedes.d@chula.ac.th

**Keywords:** informal caregiver burden, chronic illness, palliative care, bedridden elderly patient, Thailand

## Abstract

The informal family caregiver burden (IFCB) for chronically ill bedridden elderly patients (CIBEPs) is a major issue worldwide. It is a significant challenge due to the ongoing increased palliative care in the family setting; therefore, we explored the IFCB of caring for CIBEPs in Thailand. This article utilized a qualitative method, the total interpretive structural modeling (TISM) approach, with purposive sampling of thirty respondents between September and December 2020. The data were analyzed using cross-impact matrix multiplication applied to classification (MICMAC) to determine the relationship between the driving and dependence power of the enabling factors. The IFCB of the palliative care of CIBEPs was associated with primary care, nursing, extrinsic monitoring and complication prevention. The results showed that the IFCB involves taking responsibility, daily workload, follow-up caring, caring tasks, caregiving strain, financial distress, patient support, external support and caregiving strategy; thus, assistance with taking responsibility, extrinsic monitoring and follow-up care daily tasks may reduce the caregiver burden.

## 1. Introduction

In Thailand, there is a continually increasing need for palliative care post-hospitalization of elderly patients, particularly in family caregiver settings [1]. The primary care of CIBEPs is often provided by caregivers [2]. Kulkantrakorn and Suksasunee [3] reported that ALS is the second major cause of CIBEPs in Thailand as well as globally. Some studies classified that CIBEPs are related to communicable diseases and non-communicable diseases (NCDs) in the palliative care setting [4].

Caring for CIBEPs is a global issue [5], but there is a lack of data linking theory and practice [6]. Previous studies have identified that the palliative care of CIBEPs is holistic rather than clinical [7], while focusing on the different NCDs [8]. Some scholars criticized that not all IFCB in palliative care can reduce the care burden [9,10]. Previous studies suggest that the overall trajectory of caring, in particular, caregiver’s voices of subsequent the care burden [11,12].

Caregivers live with poor physical health, psychological stress and economic problems, which increase their burden [13,14]. One-third of IFCB is related to psychosocial distress [15] and poor mental health [16], with an acute health-related workload [17]. Some studies have found that the IFCB is negatively impacted by both palliative care and caregivers, such as caring tasks, fracture, monitoring, primary treatment and responsibility [6].

Despite the growing interest in the IFCB of caring for CIBEPs, few studies have explored the care burden setting. Therefore, this study investigated the palliative care tasks (primary caring treatment, performing nursing, extrinsic monitoring and complication prevention) associated with the IFCB for caring of CIBEPs (taking responsibility, daily workload, financial distress and caregiving strategy).

## 2. Context and Theoretical Background

### 2.1. Context of Bedridden Elderly Patients in Thailand

Thailand is a large country with an increasingly aging population [2,4]. According to the United Nations [18], Thailand is a completely aging society (aged ≥ 60 years), increasing from 15.6% in 2015 to 30.2% in 2035. As a result of this rapid change, caregivers in Thailand are facing challenges in caring for chronically ill patients. Furthermore, Kulkantrakorn and Suksasunee [3] point out that elderly Thai patients are at high risk of NCDs and ALS. Indeed, some 80–90% of elderly individuals have one chronic illness, and 50–77% have experienced a chronic illness more than twice [19]. Elderly Thais with chronic illness may be grouped as social, home, bedridden or dead [20]. In 2015, the Thai Ministry of Public Health reported that five million elderly patients, accounting for 21%, were living with chronic disease at home.

The central region of Thailand has a high concentration of CIBEPs, with conditions such as hypertension, diabetes, stroke and coronary artery diseases [19,21,22,23]. Suriyanrattakorn and Chang [4] defined that bedridden elderly patients have both formal (paid caregivers) and informal (unpaid caregivers) care at home. Hence, in this study we explored the care burden of unpaid caregivers providing palliative care of CIBEPs in the central region of Thailand.

### 2.2. Palliative Care

As palliative care of CIBEPs [24,25,26] is classified as prevention or treatment, we focused on life-preventing diseases (routine screening, assessment, support of care and advanced patient care) and life-threatening diseases (primary caring, performing nursing and extrinsic monitoring). Palliative care entails taking responsibility for preventing rather than treating as caregivers, which leads to a high care burden [27].

Previous studies indicated that palliative care provides a balance between the care burden and maintaining daily life [28,29]. The dominant ideal in Western palliative care emphasizes the patient’s ability to prevent and treat as caregivers [30], whereas in Asia, palliative care is performed by family caregivers [31]. These views are also supported in the recent studies of palliative care in South Asia [32], which focused on family-centered caregivers.

Palliative care is holistic [33], focusing on the quality of life of the patients, their families and caregivers. For many NCDs, palliative care has focused on prevention, treatment and life-sustaining care [34,35,36]. However, some studies concerning palliative care have specified patient types, as well as caregiver issues [37,38,39]. Palliative care of CIBEP was secondary caring burden at home [24,40]. However, caregivers in palliative care unsolved problems or unmet needs reducing the care burden [41,42,43].

### 2.3. Informal Family Caregiver Burden

Regarding caregivers and their patients as the “unit of care” is the principle of the IFCB setting [44,45]. Previous literature defined the IFCB as comprising various mental, social, physical and economic factors associated with the care burden [46]. Some studies have classified the relationship between caregiving and caregivers as perceived as negative, positive or a combination of both [12]. Caregivers provide varying care provisions throughout the day and have negative feelings toward acute health-related quality of life [42].

Numerous studies have indicated that the IFCB is associated with the caring strategy, follow-up tasks and daily workload [41,47]. Most caregivers are confronted with financial, emotional, physical and social caregiving [48,49]. Caregivers have the greatest care-induced burden, which is associated with health outcomes, support and responsibility [50,51]. Since caregivers tend to lack access to professional care and have limited care-related training, this may increase their care burden [43]. Nonetheless, the care burden can be reduced if the caregivers receive assistance with skin cleaning, follow-up care and support [8]. Caregivers include non-professional nurses such as family members, friends and paid caregivers who provide care at home [52]. Caregivers may also feel overwhelmed and have limited resources for care provision, and thus are at highest risk of a high level of care burden. Thus, it was hypothesized that the palliative care of CIBEPs is associated with an IFCB.

## 3. Methods

### 3.1. Study Design

This study utilized a qualitative TISM approach [53] to interpret the complex relationship and digraph model [54], as illustrated in Figure 1. The structural mapping of the relationships between the elements involved in informal caregiving provided a visual representation of the model [55,56]. Mathiyazhagan et al. [57] state that principle of the TISM approach is to use respondents’ experience and practical knowledge to decompose a complex system and generate a multi-level structural model.

### 3.2. Respondents and Sampling

This study was conducted in the Ayutthaya, Angthong and Pratumthani provinces in Central Thailand. The inclusion criteria were caregivers of elderly patients aged 60 years and above, those immobilized and receiving care from informal family caregivers or bedridden patients with NCDs. The respondent characteristics are provided in Table 1.

### 3.3. In-Depth Interview Questions

One-on-one in-depth interviews (Table 2) were conducted from September to December 2020 at the respondents’ home addresses, with an average interview duration of between 30 and 40 min per respondent. In-depth interviews were conducted in Thai and subsequently translated into English by the first author. The respondents were encouraged to provide a detailed description of palliative care and care burden. Each session was digitally recorded, then transcribed immediately after the interview.

### 3.4. Data Analysis

Data were analyzed using MICMAC, determining the driving (influential) power and dependence (influenced) power of each element [58]. The key factors (autonomous, linkage, dependent and independent) were defined as follows:Autonomous factors are both weak driving and dependence powers, which disconnect with others but are strongly linked with a few strong factors.Linkage factors are both strong driving and dependence powers, that is, factors act as linking (bridge) connectors with autonomous/dependent factors, which connect with independent factors.Dependent factors are less influential powers but have strong dependence power that influences the linkage/independent factors.Independent factors are strong influencing autonomous/dependent factors, which also are a strong driving power but have less dependence power.

### 3.5. Data Validity

The TIMS approach to the IFCB of caring for CIBEPs needs further validation, as it was developed with respondent views [59]. Thus, the accuracy of the elements used to build the model and the relationships of the TIMS model were cross-checked, verifying that the elements were relevant to the study context. Sushil [60] suggested checking the reachability matrix between the direct links and transitive links to validate the final transitive model of the TISM approach.

## 4. Results

The MICMAC was analyzed using responses of the thirty caregivers of CIBEPs, generating 150 codes categorized into two themes (palliative care and IFCB) as shown in Table 3 and Table 4.

### 4.1. Interpretive Logic Matrix

The respondents’ view of their experiences of caring for CIBEPs identified thirteen enabling tasks, as listed in Table 3. The logic matrix uses the symbols *I* and *j* to denote the direction nodes of ‘V’ which denotes the relationship with ‘*i*’, which leads to ‘*j*’ but ‘*j*’ does not lead to ‘*i*’. The model denoted ‘A’ enables the relationship with ‘*j*’ helping to achieve ‘*i*’, but enabling ‘*i*’ does not help to enable ‘*j*’. As elements of ‘X’ denote the relationship between both tasks, ‘*i*’ and ‘*j*’ help each other; similarly, ‘O’ represents a relationship and association with the other. The following four symbols denote the associations between elements *i* and *j* as shown in Table 5:V: element *i* will help to achieve factor *j*;A: element *j* will help to achieve factor *i*;X: element *i* and *j* will help to achieve each other;O: element *i* and *j* are unrelated.

### 4.2. Reachability Matrix

To develop a structural self-interaction matrix (SSIM), the initial reachability matrix substituted V, A, X and O as 1 and 0 as suggested by Singh and Kant [61] and shown in Table 6.

If the (*i*, *j*) entry in the SSIM is V, the (*i*, *j*) entry with the reachability matrix of 1 and the (*i*, *j*) entry become 0;

If the (*i*, *j*) entry in the SSIM is A, the (*i*, *j*) entry with the reachability matrix of 0 and the (*j*, *i*) entry become 1;

If the (*i*, *j*) entry in the SSIM is X, the (*i*, *j*) entry with the reachability matrix of 1 and the (*j*, *i*) entry also become 1;

If the (*i*, *j*) entry in the SSIM is O, the (*i*, *j*) entry with the reachability matrix of 0 and the (*j*, *i*) entry become 0.

The reachability matrix indicated the PCT and EXM values of driving power 13 and dependence power 10. The PNU values were driving power 8 and dependence power 10, and for COP the driving power 12 and dependence power was 8. The TRS had a value of driving power 12 and a dependence power of 8, while DWL had driving power 10 and dependence power 8.

The FCA driving power 11 was associated with a dependence power of 9, and CAT had a driving power of 8 and dependence power of approximately 9. The CVS driving power was 8 and dependence power was 11, while FID had a driving power equivalent to 4 and a dependence power of 7. The SUP, EXS and CGS had driving powers of 6, 4 and 6, and dependence powers of 8, 6 and 10, respectively.

### 4.3. Structural Model

The structural model was generated based on the reachability matrix, which discards the transitivity of TISM (Figure 2). The reachability set consists of one element and another which may help to achieve the antecedent set (driving and dependence power), antecedent factors (dependence power) and intersection set (reachability and antecedent set) (Table 7). The ranking level of the remaining IFCB of caring for CIBEPs was determined according to the numbers of all critical factors, which are presented in Table 8. The identified level was used to build the structural model, as shown in Figure 3.

### 4.4. TISM of MICMAC Analysis

The MICMAC analysis revealed thirteen factors enabled by the TISM approach to categorize the four clusters (autonomous, linkage, dependent and independent). The relationship of the interpretive matrix between palliative care and IFCB is illustrated in Table 9. The principles of the interpretive matrix and significant transitive links are presented in Figure 3. The driving and dependence diagram of the IFCB with CIBEPs is presented in Figure 4.

Cluster I—Autonomous factors. The MICMAC indicated that the autonomous factor for the driving power on the dependence power, thus, was positioned in cluster one, which is usually the strong driving and dependence power. As such, CGS has a weak driving power of 2 and a strong dependence power of 6, respectively.

Cluster II—Dependent factors. The dependent factors had strong driving power and weak dependence power on the IFCB of caring for CIBEPs. The critical factors of CVS and EXS had driving powers of 5 and 5, and dependence powers of 5 and 4, respectively. The TISM showed that the factor in the dependent clusters depended on the other factors, which do not support influential power.

Cluster III—Linkage factors. The MICMAC showed strong driving power and dependence power on the IFCB of caring for CIBEPs. The enabling tasks of PCT, PNU, EXM and TRS had driving powers of 9, 9, 9 and 11, and dependence powers of 9, 4, 13 and 11, respectively. The critical factors were otherwise linked because of the influential power of other factors and vice versa, which had strong driving and dependence power. Since these linking factors affect the IFCB of caring for CIBEPs, they may increase primary care treatment associated with taking responsibilities.

Cluster IV—Independent factor. The TISM illustrated that independent factors had weak driving but strong dependence power. The FID and SUP showed weak driving powers of 4 and 3, and dependence powers of 3 and 8, respectively. The critical factors of the ICB are presumed to play an important role in caring for CIBEPs, influencing all critical factors to achieve palliative care.

## 5. Discussion

### 5.1. Theory Implications

The present study provides new insights regarding the IFCB of caring for CIBEPs in Thailand, bridging the gap in the literature regarding family caregivers [8], caregiver research [62] and theory into practice [63]. This is an important theoretical contribution as it fills the gap of palliative care theory [63,64,65]. Most IFCB studies have focused on stress theory [66], caregiver identity theory [67] and single approaches to the care burden [16]. Recent articles focused on IFCB theories [43], grounded theory approach to IFCB [68] and gender role in caregiving [69]. There is a lack of data linking specific contexts, patient characteristics and the caregivers’ burden.

This study attempted to fill the gap of caregiver theory regarding IFCB in palliative care. We found that the factors contributing to IFCB were the daily workload associated with carrying out responsibilities for CIBEPs. This finding is consistent with Zubaidi et al. [40], who showed that IFCB in palliative care overload is associated with malignancy, long hours of caregiving, and symptoms. Indeed, the IFCB is associated with caregiving strain, follow-up caring and financial distress. Our article concurs with the results of Leung et al. [49], showing that IFCB in palliative care is associated with caregiver fatigue, daily activities and primary caring.

The IFCB involves various caring tasks, supporting the patient and performing nursing tasks in line with Bekdemir and Ilhan [70], who found that the informal family caring of CIBEPs comprised health constraints, activities of daily living and physical burden. Some studies identified that caregivers and patients in palliative care [29] associated with the psychological burden of caring for CIBEPs [24]. The findings also suggest that the caregiver burden based on primary care treatment, extrinsic monitoring, complication prevention and the caregiving strategy is crucially important [2,6]—the IFCB is not just a trait [5,38,40,43], which is taking responsibilities [69,71].

The IFCB of caring for CIBEPs is a multidimensional model of physical, social and financial burden [4,13,15], while some studies have suggested the use of online photovoice (OPV), which could help to reduce the care burden [72,73]. Creative methods of exploring the use of OPV with caregivers help to engage them in ways that are meaningful and investigative [74,75]. This validated approach uses strengthened theoretical contributions and enriched empirical data support. The OPV effectively engaged respondents who would then go on to meet and discuss their experiences of caregivers caring for CIBEPs.

### 5.2. Practice Implications

This study has various practical implications. First, the IFCB of caring for CIBEPs is undeniable, and thus caregivers need to be supported by complication prevention (early detection, daily add-on prevention, follow-up caring day-to-day tasks and external support) to help reduce their daily workload, caring tasks and follow-up care. The caregivers providing palliative care also need a caring strategy to help decrease the strain and care burden. Finally, ideally, long-term mental health, education, research, service and administration would be implemented to reduce the care burden associated with caring for CIBEPs. It is important to note that caregiver burden in future caring may be linked to healthcare, in planning for the caring future and caregivers’ aspects of caregiving.

### 5.3. Limitations and Further Research Directions

The present article has some limitations. First, the most obvious limitation of this article is its design as a single method of a total interpretive structural modeling approach. Second, this article is limited by the small sample size, and so may not accurately represent all groups of informal caregivers in Thailand. In-depth interviews were conducted using a fixed format questionnaire, which may have introduced bias in the data collection. Third, the findings were based on the respondents’ views, and thus cannot be generalized to other contexts. Future work should consider mixed methods, which are important to gauge the IFCB of caring for CIBEPs. Future studies should investigate different groups and clusters of IFCB, have a larger sample size, and should validate and generalize the results with empirical models.

## Figures and Tables

**Figure 1 ijerph-19-00567-f001:**
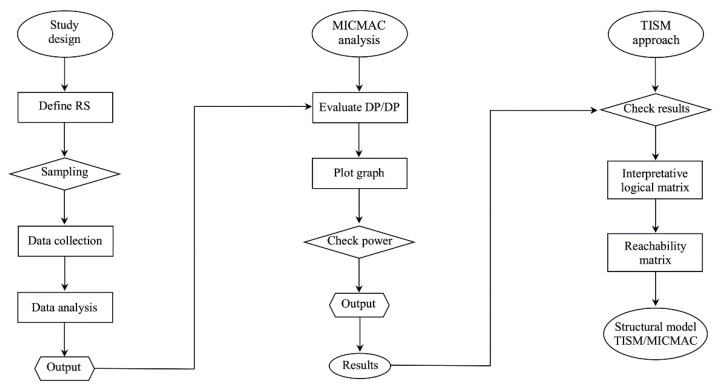
Flowchart of the TISM approach.

**Figure 2 ijerph-19-00567-f002:**
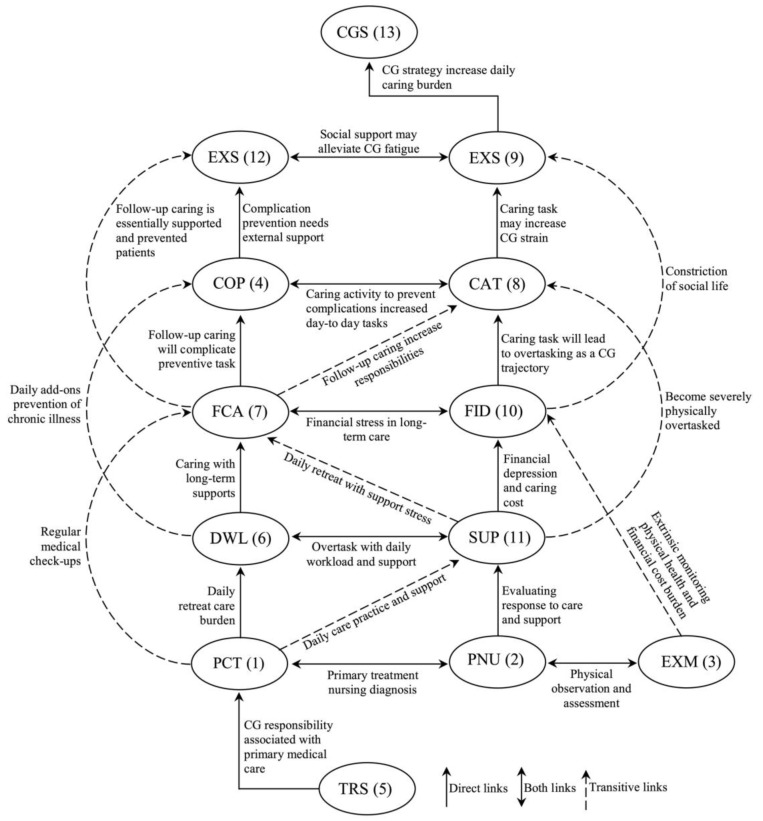
Structural model of the TISM.

**Figure 3 ijerph-19-00567-f003:**
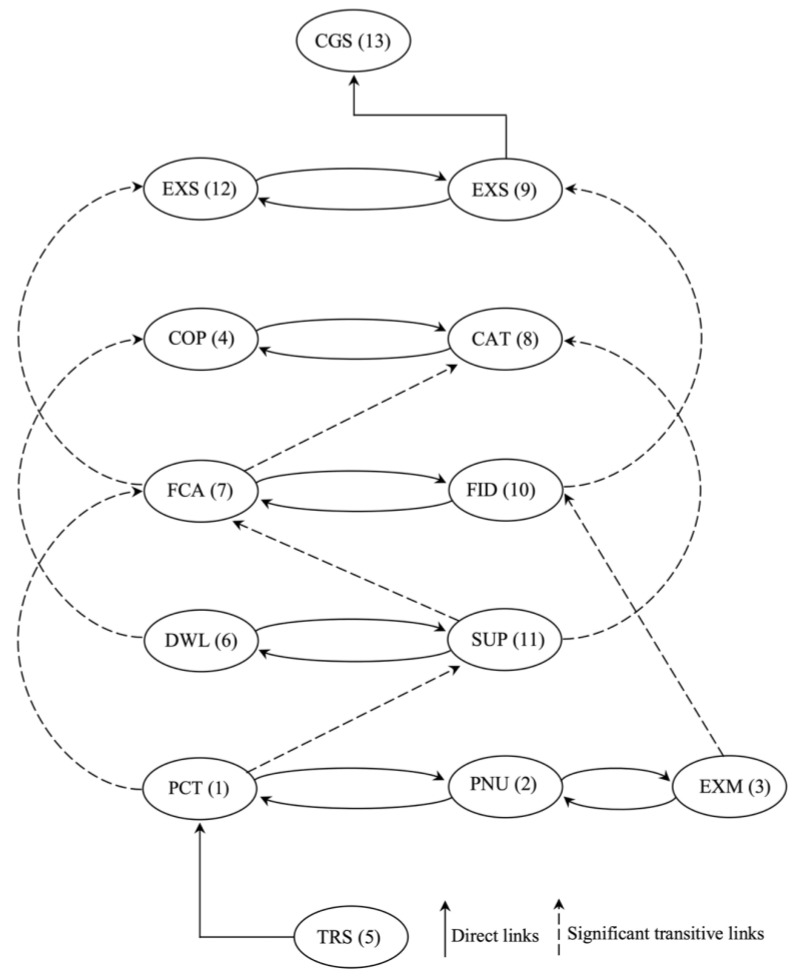
Direct with significant transitive links.

**Figure 4 ijerph-19-00567-f004:**
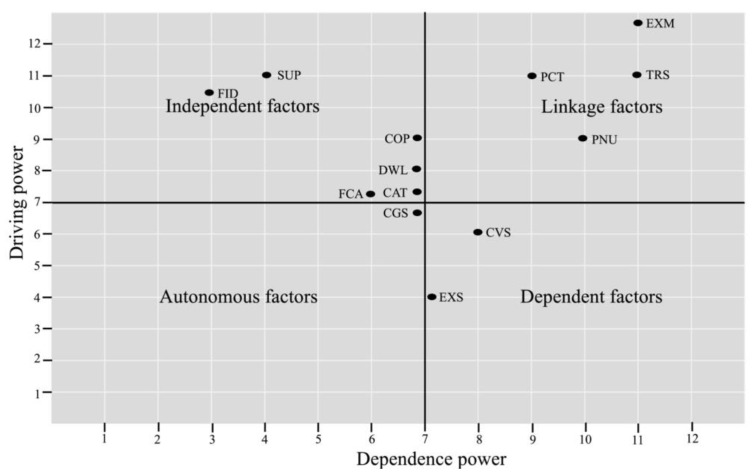
Driving power and dependence power matrix.

**Table 1 ijerph-19-00567-t001:** Respondent characteristics.

ID	CGG	CGA	MS	ED	Underlying	CD	Relationship	Income Adequacy	Medical Welfare
1	Female	61	Married	ES	Yes	2 months	Spouse	Yes	–
2	Female	44	Single	ES	Yes	8 months	Son/daughter	No	UC
3	Male	49	Married	ES	Yes	5 months	Son/daughter	Yes	UC
4	Male	53	Single	ES	Yes	5 months	Son/daughter	Yes	UC
5	Female	34	Married	ES	Yes	3 months	Son/daughter	Yes	UC
6	Female	37	Single	Diploma	Yes	6 months	Son/daughter	No	–
7	Female	53	Married	BA	Yes	10 months	Spouse	No	–
8	Female	47	Single	BA	Yes	3 months	Son/daughter	Yes	–
9	Male	70	Married	BA	Yes	1 months	Spouse	Yes	CSMBS
10	Female	36	Married	BA	Yes	8 months	Son/daughter	Yes	CSMBS
11	Female	37	Single	–	Yes	5 months	Son/daughter	No	CSMBS
12	Female	47	Married	HS	Yes	2 months	Son/daughter	Yes	CSMBS
13	Female	61	Married	Diploma	Yes	6 months	Spouse	No	UC
14	Male	59	Married	HS	Yes	2 months	Spouse	Yes	–
15	Male	59	Married	ES	Yes	3 months	Spouse	Yes	CSMBS
16	Female	44	Widow	ES	Yes	5 months	Brother/sister	Yes	CSMBS
17	Male	52	Widow	ES	Yes	2 months	Spouse	Yes	–
18	Female	49	Single	–	Yes	5 months	Spouse	Yes	–
19	Male	58	Single	–	Yes	8 months	Spouse	No	CSMBS
20	Female	58	Single	–	Yes	5 months	Spouse	No	–
21	Female	59	Single	ES	Yes	8 months	Spouse	Yes	CSMBS
22	Male	52	Widow	ES	Yes	2 months	Brother/sister	No	CSMBS
23	Female	44	Widow	HS	Yes	6 months	Son/daughter	Yes	–
24	Female	39	Single	Diploma	Yes	2 months	Son/daughter	No	–
25	Female	40	Married	ES	Yes	2 months	Son/daughter	Yes	CSMBS
26	Female	45	Married	ES	Yes	8 months	Son/daughter	Yes	–
27	Male	44	Widow	ES	Yes	2 months	Son/daughter	Yes	CSMBS
28	Male	62	Widow	–	Yes	7 months	Spouse	No	UC
29	Female	44	Single	ES	Yes	7 months	Son/daughter	Yes	CSMBS
30	Male	61	Widow	–	Yes	5 months	Spouse	No	CSMBS

Note: CGG, caregiver gender; CGA, caregiver age; MS, marital status; ED, education level; CD, caring duration; UC, universal coverage; CSMBS, civil servant medical benefit scheme.

**Table 2 ijerph-19-00567-t002:** Interview questions.

Theme	Issue	Interview Questions
Palliative care	Palliative care background Palliative care experience	Could you please describe your caregiving history from your experience with palliative care?
	Palliative care problems	What are the most important problems associated with palliative care?
	Primary caring Perform nursing Extrinsic monitoring Complication prevention	What could help you in primary caring, perform nursing, extrinsic monitoring, and complication prevention for treatment in palliative care?
Informal caregiver burden	Caregiver role Caregivers daily task Taking responsibility	What are the most important aspects important aspects of caregivers’ roles, daily activities, and responsibilities?
	Daily workload Follow-up caring Caring task	What is your daily workload, follow-up caring, and caring tasks associated with caregiver burden?
	Caregiving strain Financial distress Support of patient External support	How does your caregiving strain, financial distress, support of the patient, and external support make you feel burdened for caring?
	Caregiving strategy	Can you share with us your caregiving strategy for caring for an elderly patient in your family?

**Table 3 ijerph-19-00567-t003:** The themes and what they enabled (*n* = 30).

Theme	Enables	Acronym	Respondents Confirmed		Frequency (%)
			Time 1	Time 2	
Palliative care	Primary caring treatment	PCT (1)	√	√	28 (93.33)
	Performing nursing	PNU (2)	√	√	27 (90)
	Extrinsic monitoring	EXM (3)	√	√	30 (100)
	Complication prevention	COP (4)	√	√	30 (100)
Informal caregiver burden	Taking responsibility	TRS (5)	√	√	30 (100)
	Daily workload	DWL (6)	√	√	30 (100)
	Follow-up caring	FCA (7)	√	√	26 (86.66)
	Caring task	CAT (8)	√	√	30 (100)
	Caregiving strain	CVS (9)	√	√	25 (83.33)
	Financial distress	FID (10)	√	√	24 (80)
	Support of patient	SUP (11)	√	√	30 (100)
	External support	EXS (12)	√	√	21 (70)
	Caregiving strategy	CGS (13)	√	√	29 (96.66)

**Table 4 ijerph-19-00567-t004:** Respondents’ confirmed coding.

	(1)	(2)	(3)	(4)	(5)	(6)	(7)	(8)	(9)	(10)	(11)	(12)	(13)	(14)	(15)
1	*	*	*	*	*	*	–	*	*	*	*	*	–	*	*
2	*	*	*	*	*	*	*	*	*	*	*	*	*	*	–
3	*	*	*	*	*	*	*	*	*	*	*	*	*	*	*
4	*	*	*	*	*	*	*	*	*	*	*	*	*	*	*
5	*	*	*	*	*	*	*	*	*	*	*	*	*	*	*
6	*	*	*	*	*	*	*	*	*	*	*	*	*	*	*
7	*	*	*	–	*	*	*	*	*	–	*	*	*	*	*
8	*	*	*	*	*	*	*	*	*	*	*	*	*	*	*
9	–	*	*	*	*	*	*	*	*	*	*	*	*	*	–
10	*	*	–	*	*	*	*	–	*	*	*	–	*	*	*
11	*	*	*	*	*	*	*	*	*	*	*	*	*	*	*
12	*	*	*	–	*	*	–	–	*	*	–	*	*	*	–
13	*	*	*	*	*	*	*	*	*	*	*	*	*	*	*
	**(16)**	**(17)**	**(18)**	**(19)**	**(20)**	**(21)**	**(22)**	**(23)**	**(24)**	**(25)**	**(26)**	**(27)**	**(28)**	**(29)**	**(30)**
1	*	*	*	*	*	*	*	*	*	*	*	*	*	*	*
2	*	*	*	–	*	*	*	*	*	*	*	–	*	*	*
3	*	*	*	*	*	*	*	*	*	*	*	*	*	*	*
4	*	*	*	*	*	*	*	*	*	*	*	*	*	*	*
5	*	*	*	*	*	*	*	*	*	*	*	*	*	*	*
6	*	*	*	*	*	*	*	*	*	*	*	*	*	*	*
7	*	*	*	*	*	*	*	–	*	*	*	*	*	*	–
8	*	*	*	*	*	*	*	*	*	*	*	*	*	*	*
9	–	*	*	*	*	–	*	*	*	*	*	*	–	*	*
10	*	*	–	*	*	*	*–	*	*	*	–	*	*	*	*
11	*	*	*	*	*	*	*	*	*	*	*	*	*	*	*
12	*	–	*	–	–	*	*	–	*	*	*	*	*	*	*
13	*	*	*	*	*	–	*	*	*	*	*	*	*	*	*

Note: *, respondent confirms the coding; –, respondent rejects the coding.

**Table 5 ijerph-19-00567-t005:** Structural self-interaction matrix.

IFCB Descriptions	(1)	(2)	(3)	(4)	(5)	(6)	(7)	(8)	(9)	(10)	(11)	(12)	(13)
1	X	V	X	X	X	X	X	X	V	V	V	V	V
2	X	V	V	V	O	X	O	O	V	O	V	O	V
3	X	X	X	X	X	X	X	V	V	V	V	V	V
4	X	X	X	X	X	X	X	X	X	V	V	O	V
5	X	X	X	X	X	X	X	X	X	X	O	V	V
6	X	X	X	V	X	X	V	X	V	O	O	O	X
7	V	X	X	V	X	V	V	V	X	O	X	O	X
8	X	X	X	O	X	X	V	X	V	O	V	O	O
9	X	X	X	O	O	O	O	X	X	V	X	O	V
10	O	O	O	O	X	O	O	O	X	V	X	O	O
11	O	O	O	O	O	O	V	X	X	V	X	X	O
12	X	V	X	X	O	O	V	O	O	O	O	X	X
13	X	X	X	V	O	O	X	O	O	O	O	X	X

**Table 6 ijerph-19-00567-t006:** Initial reachability matrix.

Enables	IFCB of Caring for CIBEPs													Driving Power	Rank
	(1)	(2)	(3)	(4)	(5)	(6)	(7)	(8)	(9)	(10)	(11)	(12)	(13)		
1	1	1	1	1	1	1	1	1	1	1	1	1	1	13	1
2	1	1	1	1	0	1	0	0	1	0	1	0	1	8	5
3	1	1	1	1	1	1	1	1	1	1	1	1	1	13	1
4	1	1	1	1	1	1	1	1	1	1	1	0	1	12	2
5	1	1	1	1	1	1	1	1	1	1	0	1	1	12	2
6	1	1	1	1	1	1	1	1	1	0	0	0	1	10	4
7	1	1	1	1	1	1	1	1	1	0	1	0	1	11	3
8	1	1	1	0	1	1	1	1	1	0	1	0	0	8	5
9	1	1	1	0	0	0	0	1	1	1	1	0	1	8	5
10	0	0	0	0	1	0	0	0	1	1	1	0	0	4	7
11	0	0	0	0	0	0	1	1	1	1	1	1	0	6	6
12	0	0	1	0	1	0	0	0	0	0	0	1	1	4	7
13	1	1	1	1	0	0	1	0	0	0	0	1	1	6	6
Dependence power	10	10	10	8	7	8	9	9	11	7	8	6	10		

**Table 7 ijerph-19-00567-t007:** Level partition of each iteration.

IFCB	Reachability Set	Antecedent Set	Intersection Set	Level
1	1, 3, 5, 7, 9, 13	1, 2, 3, 4, 5, 6, 7, 8, 9, 10, 11, 12, 13	1, 3, 5, 7, 9, 13	1
2	1, 2, 3, 4, 6, 9, 11, 13	1, 2, 3, 4, 6, 9, 11, 13	1, 2, 3, 4, 6, 9, 11, 13	5
3	1, 2, 3, 4, 5, 6, 7, 8, 9, 10, 11, 12, 13	1, 2, 3, 4, 5, 6, 7, 8, 9, 10, 11, 12, 13	1, 2, 3, 4, 5, 6, 7, 8, 9, 10, 11, 12, 13	1
4	1, 2, 3, 4, 5, 6, 7, 8, 9, 10, 11, 13	1, 2, 3, 4, 5, 6, 7, 8, 9, 10, 11, 13	1, 2, 3, 4, 5, 6, 7, 8, 9, 10, 11, 13	2
5	1, 2, 3, 4, 5, 6, 7, 8, 9, 10, 12, 13	1, 2, 3, 4, 5, 6, 7, 8, 9, 10, 12, 13	1, 2, 3, 4, 5, 6, 7, 8, 9, 10, 12, 13	2
6	1, 2, 3, 4, 5, 6, 7, 8, 9, 13	1, 2, 3, 4, 5, 6, 7, 8, 9, 13	1, 2, 3, 4, 5, 6, 7, 8, 9, 13	4
7	1, 2, 3, 4, 5, 6, 7, 8, 9, 11, 13	1, 2, 3, 4, 5, 6, 7, 8, 9, 11, 13	1, 2, 3, 4, 5, 6, 7, 8, 9, 11, 13	3
8	1, 2, 3, 5, 6, 7, 8, 9, 11	1, 2, 3, 5, 6, 7, 8, 9, 11	1, 2, 3, 5, 6, 7, 8, 9, 11	5
9	1, 2, 3, 8, 9, 10, 11, 13	1, 2, 3, 8, 9, 10, 11, 13	1, 2, 3, 8, 9, 10, 11, 13	5
10	5, 9, 10, 11	5, 9, 10, 11	5, 9, 10, 11	7
11	7, 8, 9, 10, 11, 12	7, 8, 9, 10, 11, 12	7, 8, 9, 10, 11, 12	6
12	3, 5, 12, 13	3, 5, 12, 13	3, 5, 12, 13	7
13	1, 2, 3, 4, 7, 12, 13	1, 2, 3, 4, 7, 12, 13	1, 2, 3, 4, 7, 12, 13	6

**Table 8 ijerph-19-00567-t008:** Transitivity check on the reachability matrix.

IFCB Descriptions	(1)	(2)	(3)	(4)	(5)	(6)	(7)	(8)	(9)	(10)	(11)	(12)	(13)	Driving Power	Level
1	1	0	1 *	**1 ***	**1 ***	**1**	1	1	0	0	1	0	1	9	2
2	**1 ***	1	**1 ***	1	1	**1 ***	1	0	0	0	1	0	1	9	2
3	1	0	1	1	1	**1***	**1 ***	**1 ***	1	0	1	0	0	9	2
4	1	0	**1 ***	1	1	1	**1 ***	0	0	0	0	0	0	6	5
5	**1 ***	**1***	**1 ***	**1 ***	1	**1 ***	**1 ***	**1 ***	1	1	0	0	1	11	1
6	1	1	**1 ***	1	**1 ***	1	**1 ***	**1 ***	0	0	0	0	0	8	3
7	1	0	1	1	**1 ***	1	1	0	0	0	0	0	1	7	4
8	**1 ***	0	**1 ***	1	**1 ***	1	0	1	0	0	0	0	0	6	5
9	0	0	**1 ***	0	0	0	0	0	1	**1 ***	1	1	0	5	6
10	0	0	1	1	**1 ***	0	0	0	**1 ***	1	1	1	0	7	4
11	1	1	1	0	1	0	0	1	0	0	1	1	1	8	3
12	0	0	1	0	**1 ***	0	0	1	0	0	1	1	0	5	6
13	0	0	**1 ***	0	0	0	0	0	0	0	0	0	1	2	7
Dependence power	9	4	13	9	11	7	7	7	5	3	7	4	6		

Note: * Bolded text represents cells with errors corrected in the transitive relationships.

**Table 9 ijerph-19-00567-t009:** Interpretive matrix.

IFCB	(1)	(2)	(3)	(4)	(5)	(6)	(7)	(8)	(9)	(10)	(11)	(12)	(13)
1	1												
2	0	1											
3	0	1	1										
4	0	1	1	1									
5	1	1	1	1	1								
6	1	1	1	1	1	1							
7	0	1	0	0	1	1	1						
8	0	1	0	0	1	1	1	1					
9	0	0	0	0	1	1	1	1	1				
10	1	1	1	0	1	0	1	0	0	1			
11	1	1	1	1	1	1	1	1	0	0	1		
12	0	0	0	0	0	0	0	0	1	1	1	1	
13	1	1	0	0	1	0	1	0	0	0	0	0	1

## Data Availability

Data will be available upon reasonable request to the corresponding author.

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
