# Peer review of "Listening to Caregivers’ Voices: The Informal Family Caregiver Burden of Caring for Chronically Ill Bedridden Elderly Patients"

_ijerph, 2022, doi:10.3390/ijerph19010567_

Round 1
Reviewer 1 Report
congrats for your meaningful work and see my detailed feedback.
Title of the work: Listening to Caregivers’ Voices: The Informal Family Caregive Burden of Caring for Bedridden Elderly Patients with Chronic Illness in Thailand
Dear researcher(s), you are addressing an important and meaningful gap. Your paper is well-written and it has some important results, and if you edit your paper it can be much more effective. Here some humble suggestions to improve the paper, I would do the following to strengthen the paper. I have enjoyed reading the paper and am looking forward to seeing the paper published. You could increase the effect of your paper with some more recent studies suggested below or any other studies and not using the suggested ones.
Main points:
- Title: good and
- You may consider shorten the title because more and more journals are asking for manuscripts with lesss number of total words. If the title is brief, comprehensive the readers and researchers will be more likely to benefit from it more. However, you do not have to shorten it.
- You may consider “The Informal Family Caregivers’ Voices for Bedridden Elderly Patients with Chronic Illness”
- Abstract and keywords clear and comprehensive: good and
- You can clearly say, the results are…. For the moment it is not easy to see the results.
- You can provide abbreviation of ALS for “amyotrophic lateral sclerosis”
- Overall language:
- The language is quite clear and well-written. You could use an active language for your future papers throughout the paper since an active language seems to be more effective. And more and more researchers go with an active language. However, you do not have to change for this paper- just a suggestion for your future work and I know some journals asking for a passive language.
Tanhan, A., & Strack, R. W. (2020). Online photovoice to explore and advocate for Muslim biopsychosocial spiritual wellbeing and issues: Ecological systems theory and ally development. Current Psychology, 39(6), 2010-2025. https://doi.org/10.1007/s12144-020-00692-6
Tanhan, A., Arslan, G., Yavuz, K. F., Young, J. S., Çiçek, Ä°., Akkurt, M. N., Ulus, Ä°. Ç.,Görünmek, E. T., Demir, R., Kürker, F., Çelik, C., Akça, M. Åž., Ünverdi, B., Ertürk, H., & Allen, K. (2021). A constructive understanding of mental health facilitators and barriers through Online Photovoice (OPV) during COVID-19. ESAM Ekonomik ve Sosyal AraÅŸtırmalar Dergisi, 2(2), 214-249. https://dergipark.org.tr/en/pub/esamdergisi/issue/64932/956618
- Length of paragraph : good and you can check the paper and make sure every paragraph is not more than 5 sentences. The best is to stick with 3 to 5 sentences. And you can add some subtitles to be more organized and short sections. Please see the suggested papers above.
- Introduction: good and
- Better to use more comman words “avows a holistic”
- You can put subheadings to make it more effective, see suggested studies (e.g., purpose of the study, BEPCI in in the world, BEPCI in Thailand)
- Intro can be a little more extended, see suggested papers and yet do not have to
- Thoroughness of the literature review: can support with some recent studies yet do not have to. This will increase the effect of the paper and the journal.
- Clarity of the description of the Theoretical Framework (TF): You do not have to construct a subsection called theoretical framework because you have enough explanation about which theories/perspectives have shaped your research. However, if you want to make your paper more effective I would construct a subsection called theoretical framework just before you start the method section and briefly explains.And if you would like to learn what makes having a theoretical framework very necessary, you can read theoretical framework section in this recent paper. Having a clear theoretical framework will make your study much more effective especially for a long run. And this will also benefit our journal that help all of us to contribute to science very quickly.
- Briefly providing research design/method: good and
- “December 2019 at the respondents….” Can you chech the intro “December 2020…” I cannot remember the dtails but make sure to e consistentn
- Clearly providing research questions and/or purpose: good and can be more effective you put a subheading
- Choice of research method: very appropriate
- Clarity of presentation of research method : well-written
- Appropriateness of procedures chosen for data collection and analysis: well-written
- Relevance of data obtained in view of the purpose of the research: well-written
- Discussion of the results and their significance: well-written
- Soundness of conclusions in relation to data presented: well-written.
- Limitation: well-written.
- Implication: good and
- you can increase the effect of your paper by constructing a new section entitled “implication” for clear and brief suggestions in at least two or three of the following most important to you mental health, education, research, administrators, services, etc.: see suggested papers for implications for specific sections
- I would strongly suggest you to call future researchers to use TISM and another quite new method Online Photovoice (OPV) to conduct research on the same or similar topics: understanding experience of IFCB and BEPCI. They can use OPV as one of the most recent and effective innovative qualitative research methods. OPV gives opportunities to the participants to express their own experience with as little manipulation as possible if at all, compared to traditional quantitative methods. As researchers one of our responsibilities is to inform others about recent and effective methods, which will increase the effect of your paper and the journal. Future researchers can conduct only qualitative or mixed method to see if OPV. And educators/trainers etc. also can use OPV for experiential activities to increase group and organizational synergy. Please see suggested papers if you wish to do so.
- And the first OPV paper published in Current Psychology and it has been one of the most effective papers in the journal and it seems it will get even much more effect in the following years
- Figure/tables: very good
- In-text reference: check based on the journal style
- References:
- You could increase the effect of your paper with some more recent studies
- Please use the following link to include all available doi numbers https://doi.crossref.org/simpleTextQuery simply include your reference one or more than one at a time and submit it. Then you should get all doi numbers if a manuscript has it.
In sum, you are addressing an important and meaningful gap. Your paper has some important results, and if you edit your paper based on all or some of the humble suggestions above, it can be much more effective. I have enjoyed reading the paper and am looking forward to seeing the paper published. You could increase the effect of your paper with some more recent studies suggested above or any other studies and not using the suggested ones.
Author Response
All responses to the reviewers’ comments is attached file.

Reviewer 2 Report
Dear Authors,
Thank you for the opportunity to read your work. It is well written manuscript with important findings.
There are however some aspects that could be corrected:
- abbreviations used for the firts time should be explained;
- data given in Table 1 - would it be possible to provide it in a short version, combining all and give sums?
- manuscript contains too many tables/figures. I am aware that method used implies necessity of the number of tables, but maybe the authors could consider to reduce their number.
Generally, it is not easy to follow your results, it would be good to write them in a more descriptive way.
Author Response
Letter in response to the reviewers’ comments
Dear Reviewer
Thank you for reviewing the manuscript entitled “Listening to Caregivers’ Voices: The Informal Family Caregiver Burden of Caring for Chronically Ill Bedridden Elderly Patients”. We have now addressed all reviewers’ and the new changes in the manuscript are tracked changes in the text. In our revised version, we have now uploaded both track changes and clean-copy below. We believe the manuscript is now stronger and clearer.
Sincerely,
The authors
Response to reviewer #2
Point 1:
Thank you for the opportunity to read your work. It is well written manuscript with important findings.
Response 1:
Thank you for taking the time to carefully review our manuscript. Your comment and for pointing out that we have to clarify and improve our paper strong and clear.
Point 2:
There are however some aspects that could be corrected: abbreviations used for the first time should be explained; data given in Table 1 - would it be possible to provide it in a short version, combining all and give sums?
Response 2:
Thank you for your comment concerning our abbreviations and Table 1. We have now clarified, including word, term and abbreviations as requested. In our revised version, we have now modified Table 1.
Point 3:
Manuscript contains too many tables/figures. I am aware that method used implies necessity of the number of tables, but maybe the authors could consider to reduce their number.
Response 3:
Thank you for your comments. We have now expanded and modified the tables/figures as suggested. First, all tables are based on TISM approach. Second, all figures are based MICMAC results. Then, we have also reinforced the importance of testing the TISM on MICMAC model.
Point 4:
Generally, it is not easy to follow your results, it would be good to write them in a more descriptive way.
Response 4:
Thank you for your thoughtful comment. We have now explained the analysis, including both TISM and MICMAC findings.
Reviewer 3 Report
Dear authors, both the methodology and the subject of the study address a very important issue. Just a few issues.
- Were the interviews the same for all participants? Were open or closed questions asked?
- Was it the same interviewer all the time? this could be accounted for by biases.
- The age range of the sample goes from 34 to 70, according to table 1. Isn't it a very large range to which differences may be due?
- Did the care time influence the results?
- Are there the same number of men as women in the study?
Great the implications and consideration of mixed methods as well as the limitations that the study presents.
Congratulations on the job.
All the best.
Author Response
Letter in response to the reviewers’ comments
Dear Reviewer
Thank you for reviewing the manuscript entitled “Listening to Caregivers’ Voices: The Informal Family Caregiver Burden of Caring for Chronically Ill Bedridden Elderly Patients”. We have now addressed all reviewers’ and the new changes in the manuscript are tracked changes in the text. In our revised version, we have now uploaded both track changes and clean-copy below. We believe the manuscript is now stronger and clearer.
Sincerely,
The authors
Response to reviewer #3
Point 1:
Dear authors, both the methodology and the subject of the study address a very important issue. Just a few issues.
Response 1:
Thank you for taking the time to carefully review our manuscript. Your comment and for pointing out that we have to clarify and improve our paper strong and clear.
Point 2:
Were the interviews the same for all participants? Were open or closed questions asked?
Response 2:
Thank you for your thoughtful question. We interview all respondents are same time and interview questions.
Point 3:
The age range of the sample goes from 34 to 70, according to table 1. Isn't it a very large range to which differences may be due?
Response 3:
Thank you for your constructive question. The age of respondents are not our data collection criteria, because we collected the representative of caregiver care patients. We found that caregiver aged ranged from 34 to 70 years.
Point 4:
Did the care time influence the results?
Response 4:
Thank you for your question. Care time of the patient is influencing the results. This is because caregivers who providing care the patient in difference age, education, family status (economic/finance), which influences to caregiver burden.
Point 5:
Are there the same number of men as women in the study?
Response 5:
Thank you for your insightful question. Our data were collected from a caregiver who provided caring patient. On our criteria in data collecting is based on caregivers caring patients. At that time, we found that women more than men caring the patients at home.
Point 6:
Great the implications and consideration of mixed methods as well as the limitations that the study presents.
Response 6:
Thank you for your constructive suggestion.
Round 2
Reviewer 1 Report
Congrats for your meaningful editing. The paper is much better now.
Reviewer 2 Report
I am satisfied with corrections
Reviewer 3 Report
Thanks for your answers.
Best wishes.